# Validity and reliability of the Serbian COVID Stress Scales

Marija Milic[1], Jelena Dotlic[2,3], Geoffrey S. Rachor[4], Gordon J. G. Asmundson[4], Bojan Joksimovic[5], Jasmina Stevanovic[1], Dragoslav Lazic[6], Zorica Stanojevic Ristic[7], Jelena Subaric Filimonovic[1], Nikoleta Radenkovic[1], Milica Cakic[2], Tatjana Gazibara[8]*

1 Department of Epidemiology, Faculty of Medicine, University of Pristina Temporarily Seated in Kosovska Mitrovica, Kosovska Mitrovica, Kosovo, Serbia, 2 Clinic for Obstetrics and Gynecology, Clinical Center of Serbia, University of Belgrade, Belgrade, Serbia, 3 Faculty of Medicine, University of Belgrade, Belgrade, Serbia, 4 Department of Psychology, University of Regina, Regina, Canada, 5 Faculty of Medicine, University of East Sarajevo, Foca, Republic of Srpska Bosnia and Herzegovina, 6 Department of Dentistry, Faculty of Medicine, University of Pristina Temporarily Seated in Kosovska Mitrovica, Kosovska Mitrovica, Kosovo, Serbia, 7 Department of Pharmacology, Faculty of Medicine, University of Pristina Temporarily Seated in Kosovska Mitrovica, Kosovska Mitrovica, Kosovo, Serbia, 8 Institute of Epidemiology, Faculty of Medicine, University of Belgrade, Belgrade, Serbia

☉ These authors contributed equally to this work.
* tatjanagazibara@yahoo.com

**Data Availability Statement:** All relevant data are within the manuscript and its Supporting Information files.

## Abstract

This study aimed to generate a linguistic equivalent of the COVID Stress Scales (CSS) in the Serbian language and examine its psychometric characteristics. Data were collected from September to December 2020 among the general population of three cities in Republic of Serbia and Republic of Srpska, countries where the Serbian language is spoken. Participants completed a socio-demographic questionnaire, followed by the CSS and Perceived Stress Scale (PSS). The CSS was validated using the standard methodology (i.e., forward and backward translations, pilot testing). The reliability of the Serbian CSS was assessed using Cronbach's alpha and McDonald's omega coefficients and convergent validity was evaluated by correlating the CSS with PSS. Confirmatory factor analysis was performed to examine the construct validity of the Serbian CSS. This study included 961 persons (52.8% males and 47.2% females). The Cronbach's alpha coefficient of the Serbian CSS was 0.964 and McDonald's omega was 0.964. The Serbian CSS with 36 items and a six-factorial structure showed a measurement model with a satisfactory fit for our population (CMIN/DF = 4.391; GFI = 0.991; RMSEA = 0.025). The CSS total and all domain scores significantly positively correlated with PSS total score. The Serbian version of the CSS is a valid and reliable questionnaire that can be used in assessing COVID-19-related distress experienced by Serbian speaking people during the COVID-19 pandemic as well as future epidemics and pandemics.

## Introduction

The COVID-19 pandemic has impacted the psychological health of the global population for more than one year. The first case of COVID-19 in the Republic of Serbia was identified on

**Funding:** The authors received no specific funding for this work.

**Competing interests:** The authors have declared that no competing interests exist.

March 6, 2020. Until the end of June 2021, more than 700,000 confirmed cases of COVID-19 were registered, and more than 7,000 people died due to COVID-19-related complications. The dynamic of the epidemic over the past 15 months in Serbia could be divided in distinctive three waves (March to end-of-May 2020; July to the end of August 2020; October 2020 to the mid-May 2021, when the largest number of people caught the novel coronavirus and died as a result) [1].

Pandemic has led to substantial disruptions in health care delivery [2, 3], daily routines (e.g., mandatory use of face masks) [4, 5], and mental well-being [6, 7]. While mental health burden was first recognized among health care workers, due to excessive workload, deterioration in psychological well-being has also been observed among the general populations [8–11].

Researchers have demonstrated that stress and anxiety are common during epidemics and pandemics [12, 13]. The present pandemic is different from previous pandemics in recent decades in that everyday functioning for the global population has extraordinarily changed. For example, in addition to mandatory use of facemasks, governments have implemented social distancing protocols and closures of non-essential businesses, and several workplaces have shifted to having employees work remotely [4, 5]. To accurately define mental health challenges that people encounter daily during the COVID-19 pandemic, particularly the stress associated with potential contagion as well as the economic burden and potential xenophobia due to the closing of borders and reduction of international travel, there is an increased need for valid instruments to measure COVID-19-related distress [6, 7].

Recent studies suggested that the effect of COVID-19 on mental health differed between countries that applied various strategies to prevent and control the pandemic [14, 16]. Although an increase in frequency of manifested psychological symptoms during the pandemic was observed worldwide, especially symptoms of depression and anxiety, the impact on mental health was stronger in countries where the epidemiological situation was worse [14]. For example, in the United Kingdom, where lockdown was came into force somewhat later than in other countries in the European Union and funding of preventive measures was ten times lower compared to Germany, a more severe direct impact of COVID-19 on health, financial situation and families. By contrast, people in Germany were less optimistic regarding the end of the epidemic, but more worried about their life [15]. Another study that compared people in Poland and China found that less strict demand for face masks use in Poland compared to China was associated with more intense anxiety, depression and stress, as well as physical symptoms related to the COVID-19 infection [16].

Over the first wave of the pandemic, both Republic of Serbia and Republic of Srpska (Bosnia and Herzegovina) entered a lockdown, although residents who were living abroad were allowed to enter the country. During lockdown, people aged 65 and above were not allowed to leave their homes and were allocated a time slot once a week to shop for groceries, while curfew for all citizens lasted throughout the entire weekends. Reopening was gradual along with the improvement of the epidemiological situation [17]. Over the following months and the epidemic waves, various preventive and control measures were in place (working/schooling from home, closure of public spaces and ban of celebrations and gatherings of more than 5 people, reduction of working hours of cafes and restaurants, mandatory use of face masks inside and outside, social distancing, etc) except lockdown. The borders remained open during second and third epidemic wave although incoming people were requested to present certificate of negative PCR coronavirus test [17]. In Serbia, a national psychological telephone support service during lockdown. One quarter of people who used this service was older than 70 years, 63.6% were women, and one third reported having symptoms of anxiety and feelings of tension during the COVID-19 epidemic [18].

The COVID Stress Scales (CSS) were designed at the beginning of the COVID-19 pandemic in English language [19]. The CSS measures various facets of COVID-19-related distress,

including fears about the dangerousness of COVID-19 and of contamination, socio-economic concerns, xenophobic attitudes, traumatic stress, compulsive checking and reassurance seeking symptoms. Based on data from literature and remarks from experts in the field of anxiety, the CSS included rigorous psychometric testing which resulted in reduction of the initial pool of items. The final version of the CSS is compact but versatile, which may allow researchers and practitioners to further adapt the scale in future pandemics [19].

The CSS has already been translated into several languages (e.g., psychometric properties have been assessed in Persian language) and has contributed to a growing body of evidence regarding the impacts of COVID-19-related distress and COVID stress syndrome [20, 21]. Because of the similar circumstances and public health measures across multiple countries, use of the CSS in various cultures could provide more nuanced insights into stress stemming from the COVID-19 pandemic. The purpose of this study was to generate a linguistic equivalent of the CSS in the Serbian language and examine its psychometric characteristics.

## Material and methods

### Setting

This study was carried out in two regions/countries where Serbian language is spoken: Republic of Serbia and Republic of Srpska (Bosnia and Herzegovina). Data were collected over the course of four months (September–December) in 2020. The recruitment of study participants was carried out in four randomly selected cities, three in Republic of Serbia (Belgrade, Kragujevac and Kosovska Mitrovica) and one in Republic of Srpska—Bosnia and Herzegovina (Foca).

We printed the names of all 29 official administrative cities in Serbia and 8 in Republic of Srpska on separate slips of paper folded them for blinding and placed them in a in a non-transparent container. In table of random numbers, we hit number nine and consequently draw from the container every ninth slip of paper with city names. In Republic of Serbia there are 5 regions (Belgrade, Vojvodina Region, Sumadija and Western Serbia region, Southern and Eastern Serbia region and Kosovo and Metohija region) with 197 municipalities [22]. Rural and suburban regions are divided and clustered into municipalities according to population size and geographical localizations. On the other hand, towns and cities form municipalities based on the number of residents. Smaller towns and cities are considered as one municipality while larger cities are divided into different municipalities for easier administration. Therefore, in Serbia there are 17 municipalities in the Belgrade region, 45 in the region of Vojvodina, 53 in the region of Sumadija and Western Serbia, 53 in the region of Southern and Eastern Serbia 29 in the region of Kosovo and Metohija [22]. However, because some cities had just one and others multiple municipalities, to overcome the discrepancies in the population size, we decided to distribute the questionnaires in only one randomly chosen municipality per city. We printed the names of all municipalities on separate slips of paper, folded them for blinding and placed them in one non-transparent container per city after which we chose one municipality from every container.

### Selection of study participants

All people who came to the chosen municipality office headquarters to engage in regular administrative business were invited to participate in the study. In Serbia and Republic of Srpska (Bosnia and Herzegovina), civil administration affairs are just starting to function online. The vast majority of people who require any administration related to taxation or are in need of official certificates (birth, death, marriage, divorce, etc) and other paperwork are required to come in person to the municipality office. Thus, many people visit their municipality office

quite often. This approach enabled us to select people at random and include individuals of different socio-demographic backgrounds.

The inclusion criteria were: being ≥18 years, being of Serbian nationality and speaking Serbian language fluently. The exclusion criteria were: reporting psychiatric disorders previously diagnosed by a physician that could evidently negatively impact self-perception and comprehension (all F-coded diagnoses according to the International Classification of Diseases, 10th revision) and providing less than 90% of answers. In total, we approached 1,347 persons out of which approximately 71% fulfilled the study criteria. We excluded 19 individuals due to confirmed psychiatric and 367 due to not fulfilling 90% of the questionnaire. The final sample included approximately 0.36% of the population of the appraised cities.

The sample size is often dependent on the length of the questionnaire. Given that the CSS has 36 items, we aimed to minimize bias arising from the number of observations in order to perform a robust confirmatory factor analysis. Thus we opted for participant-to-item ratio of 25:1 (i.e. minimum sample size 36 x 25 = 900). Although some authors recommend that the participant-to-item ratio be at a minimum 5:1 [23], larger sample sizes could provide more meaningful factor loadings and factors and yield more generalizable results.

The Ethics Committee of the Faculty of Medicine, University of Pristina temporarily seated in Kosovska Mitrovica approved the study (approval no. 10-1285/1).

## Instruments

All instruments applied in the study were self-administered in paper-and-pencil form. At any moment, at least one researcher was present to provide all necessary explanations. Socio-demographic characteristics, lifestyle and habits as well as short medical history data were gathered from all study participants using a general demographics questionnaire.

COVID Stress Scales (CSS). The CSS was constructed to better understand and assess COVID-19-related distress and health-related anxiety during times of pandemic [19]. The scale has 36 items which are grouped into six domains, including 1. danger (DAN), 2. socio-economic consequences (SEC), 3. xenophobia (XEN), 4. contamination (CON), 5. traumatic stress symptoms (TSS) and 6. compulsive checking and reassurance seeking (CHE). Items are rated on a 5-point scale ranging from 0 (not at all) to 4 (extremely) for fear-related items in the DAN, SEC, XEN, and CON domains. The TSS and CHE domains are also rated from 0 (never) to 4 (almost always), however, responses are related to frequency rather than intensity. The scores for each of the six domains are calculated as the sum of ratings for each item in that domain. Higher scores indicate more intense or more frequent perceptions. The composite CSS score (i.e., total score) is the sum of all six domain scores and ranges from 0 (i.e., low COVID-19-related distress) to 144 (i.e., severe COVID-19-related distress).

The Perceived Stress Scale (PSS) is the most frequently used scale to measure perceived stress [24]. The scale assesses the degree to which individuals evaluate their life circumstances and situations as stressful. The PSS includes not only items regarding the symptoms of psychological distress (e.g., anxiety and depression) but also encompasses perception of unpredictability, uncontrollability, and sense of overload, making it sensitive to both current life conditions and expectations for future. The PSS consisted of 10 items rated from 0 (never) to 4 (very often) [24]. The PSS scores are obtained by reversing responses to the positive items (i.e., items 4, 5, 7 and 8) and then adding the item-level scores across all items. Higher total scores indicate greater levels of stress. In the available literature norms for the original PSS are based on US population and encompass the mean +/- standard deviation values of PSS total score according to gender, age groups and race [24]. The PSS has previously been translated and validated in the Serbian population [25].

## Translation of the COVID Stress Scales

The translation and use of the CSS to Serbian was approved by the authors of the scale. We first translated the CSS from English to Serbian (forward translation). The translations were performed by two independent translators who were fluent in English. The two forward translated versions were analyzed. A third professional English translator, who was blinded to the original version of the questionnaire, performed back translation from Serbian to English language. Afterwards, all translators discussed the three translations to identify discrepancies.

After some modifications and consensus between the translators (S1 File), we generated the final version of the CSS in Serbian (S1 File). This version was tested on 10 adults, for understanding and additional comments about clarity. No appreciable remarks were noted in this process. The final version was then emailed to the questionnaire authors for approval. Upon discussion about the Serbian translation with the authors of the CSS and minor adjustments, we were granted the permission to test the psychometric properties of the Serbian CSS. The Serbian translation (in both Cyrillic [official alphabet] and Latin alphabet) can be found on the official CSS website https://coronaphobia.org/professional-resources/.

## Statistical analysis

We used mean value and standard deviation (SD) of demographics as well as CSS and PSS total scores to characterize the study sample. To describe the CSS we also analyzed minimal and maximal values, skewness, and kurtosis for each item. Data analyses were performed using the SPSS 20.0 statistical software package (SPSS Inc., Chicago, IL, U.S.A.) and the JASP version 0.14.1 (http://jasp-stat.org). AMOS software version 18.0 was used to conduct CFA.

The scale internal consistency and reliability was evaluated by examining Cronbach's alpha and McDonald's omega coefficients [25, 26]. The alpha coefficient is a common indicator of whether the scale is internally consistent. The omega coefficient represents composite reliability i.e. construct reliability and is therefore recommended to estimate "matter-of-fact" reliability of the scale, as the coefficient uses the strength of association between items and constructs as well as item-specific measurement errors. Adequate levels for both coefficients are >0.7 [27]. Discriminating characteristics of the scale items were examined by means of Corrected Item–Total Correlation (CI–TC). These values take into account the relationships of one item with the score of the remaining items in the scale. Items with CI–TC ≥0.40 are regarded as suitable as they are consistent with the averaged results of other items. Finally, we used Hotelling's t-squared test to assess the existence of a meaningful difference between mean score values of all CSS items together and the hypothetical case in which items have equal scores [28, 29].

Initially, to define whether our data were suitable to run factor analysis, we observed Kaiser-Meyer-Olkin measure of sampling adequacy and Bartlett's test of sphericity. Bearing in mind that the factorial structure of the CSS may differ between populations [19, 20], we decided to evaluate both the six and five-factor structure of the Serbian CSS using the confirmatory factor analysis (CFA). We examined several parameters on the CFA for appropriate fit. First, the $\chi^2$ test with degrees of freedom (df) allows reporting potential difference between the observed and the expected covariance matrices. Values of $\chi^2$/df (CMIN/DF) below 5.0 and a p-value of greater than 0.05 confirm the set (i.e., original model fit). However, evaluation of p-value in small and large samples is less reliable. Second, the root mean square error of approximation (RMSEA) avoids issues of sample size by analyzing the discrepancy between the hypothesized model with optimal estimates and the population covariance matrices. Adequate RMSEA value is below 0.08. Third, concurrence of hypothesized model with the observed covariance matrix is assessed by the goodness of fit index (GFI) and the adjusted goodness of fit index (AGFI). Fourth, the comparative fit index (CFI) shows the discrepancy between the

data and the hypothesized model, while adjusting for the issues of sample size. Fifth, non-normed fit index (NNFI) is based on a comparison of the $\chi^2$ of the implied matrix with that of a null model in which all observed variables are uncorrelated. Values of the latter four indices above 0.9 indicate acceptable model fit [28, 29].

Criterion validity was tested using the Spearman's correlation of the CSS domains and total scores with PSS total score. We also compared the CSS against PSS using the receiver operating characteristics (ROC) analysis to define potential cut-off scores of CSS according to PSS for higher levels of stress.

## Results

### Description of the study sample

A total of 961 participants were included in the analyses. The mean age of participants was 38.2 (SD = 14.1) years, of which 507 (52.8%) identified as males and 454 (47.2%) as females. The majority of participants reported having secondary education (54%), being permanently employed (50.8%), and having no chronic illnesses (59.8%). The largest proportion of participants were married (45.3%) and lived, on average, with 3.6 (SD = 1.6) household members. In our sample, 25% (n = 240) confirmed having contact with a COVID-19 positive person, but only 18.4% (n = 177) respondents were tested for COVID-19 due to typical symptoms, out of which 52 (3.9%) had COVID 19 at some point. To seek information about COVID-19, study participants most often used non-medical (58.9%) sources like media, internet and friends.

### COVID Stress Scales scores

Average CSS item scores are presented in Table 1. Data were not normally distributed, but skewness and kurtosis were appropriate.

In the Serbian population, scores ranged from 0 to 4 for all items; however, all items marked as four by participants occurred significantly less often compared to other ratings. The mean total CSS was 35.4 (SD = 25.9), with a range from 0 to 144, suggesting moderately high overall stress due to COVID-19. The highest average score per domain was achieved for DAN scale and the lowest for TSS scale (Table 2).

### Internal consistency

The Cronbach's alpha coefficient of the Serbian CSS was 0.964 (CI 0.960 to 0.967) and McDonald's omega was 0.964 (CI 0.961 to 0.967). When based on standardized items, the alpha coefficient was 0.965. We observed that Cronbach's alpha was adequate for each CSS domain (Table 2). The values of Cronbach's alpha coefficient if-item-deleted were appropriate for all items. All coefficients ranged from 0.962 to 0.964. However, the highest coefficients were observed when items 3, 33, and 36 were deleted (Table 1). Cronbach's alpha values for the CSS domains were also all above 0.8 (Table 2).

Based on the Hotelling's t-squared test, there was a significant difference ($HT^2 = 2255.146$; F = 62.151; p = 0.001) between item scores. The highest average score was achieved for items 22 (CON) and 3 (DAN), while item 30 (TSS) had the lowest average score. The values of the CI-TC coefficient for the Serbian CSS were higher than 0.40 for all items, with the lowest of 0.463 for item #3 (DAN).

### Construct validity

The sampling adequacy according to the KMO criteria was 0.954 and Bartlett's test of sphericity showed the probability value of p<0.001, suggesting that data are suitable for factor

**Table 1. Average COVID Stress Scales item scores and their reliability parameters.**

| Items | Mean | SD | Skew | Kurt | CI—TC | Alpha if item deleted | Omega if item deleted |
|---|---|---|---|---|---|---|---|
| 1. I am worried about catching the virus | 1.48 | 1.05 | 0.27 | -0.48 | 0.634 | 0.963 | 0.963 |
| 2. I am worried that I can't keep my family safe from the virus | 1.84 | 1.17 | 0.08 | -0.85 | 0.560 | 0.963 | 0.964 |
| 3. I am worried that our healthcare system won't be able to protect my loved ones | 1.84 | 1.24 | 0.09 | -1.01 | 0.463 | 0.964 | 0.964 |
| 4. I am worried our healthcare system is unable to keep me safe from the virus | 1.65 | 1.16 | 0.19 | -0.81 | 0.517 | 0.963 | 0.964 |
| 5. I am worried that basic hygiene is not enough to keep me safe from the virus | 1.46 | 1.14 | 0.38 | -0.72 | 0.604 | 0.963 | 0.963 |
| 6. I am worried that social distancing is not enough to keep me safe from the virus | 1.39 | 1.12 | 0.38 | -0.65 | 0.599 | 0.963 | 0.963 |
| 7. I am worried about grocery stores running out of food | 0.68 | 1.02 | 1.46 | 1.34 | 0.678 | 0.963 | 0.963 |
| 8. I am worried that grocery stores will close down | 0.66 | 1.03 | 1.64 | 1.97 | 0.663 | 0.963 | 0.963 |
| 9. I am worried about grocery stores running out of cleaning or disinfectant supplies | 0.70 | 1.01 | 1.44 | 1.44 | 0.707 | 0.962 | 0.963 |
| 10. I am worried about grocery stores running out of cold or flu remedies | 0.93 | 1.13 | 0.99 | -0.05 | 0.709 | 0.962 | 0.962 |
| 11. I am worried about grocery stores running out of water | 0.66 | 1.05 | 1.54 | 1.47 | 0.652 | 0.963 | 0.963 |
| 12. I am worried about pharmacies running out of prescription medicines | 0.91 | 1.18 | 1.15 | 0.29 | 0.691 | 0.962 | 0.963 |
| 13. I am worried that foreigners are spreading the virus in my country | 1.29 | 1.26 | 0.64 | -0.73 | 0.615 | 0.963 | 0.963 |
| 14. If I went to a restaurant that specialized in foreign foods, I'd be worried about catching the virus | 1.15 | 1.19 | 0.78 | -0.36 | 0.691 | 0.962 | 0.962 |
| 15. I am worried about coming into contact with foreigners because they might have the virus | 1.23 | 1.23 | 0.72 | -0.50 | 0.671 | 0.963 | 0.963 |
| 16. If I met a person from a foreign country, I'd be worried that they might have the virus | 1.29 | 1.20 | 0.66 | -0.51 | 0.683 | 0.962 | 0.963 |
| 17. If I was in an elevator with a group of foreigners, I'd be worried that they're infected with the virus | 1.51 | 1.24 | 0.45 | -0.88 | 0.659 | 0.963 | 0.963 |
| 18. I am worried that foreigners are spreading the virus because they're not as clean as we are | 1.01 | 1.21 | 0.94 | -0.26 | 0.654 | 0.963 | 0.963 |
| 19. I am worried that if I touched something in a public space, I would catch the virus | 1.09 | 1.15 | 0.87 | -0.11 | 0.746 | 0.962 | 0.962 |
| 20. I am worried that if someone coughed or sneezed near me, I would catch the virus | 1.12 | 1.07 | 0.79 | -0.06 | 0.693 | 0.962 | 0.963 |
| 21. I am worried that people around me will infect me with the virus | 1.01 | 1.03 | 0.92 | 0.29 | 0.754 | 0.962 | 0.962 |
| 22. I am worried about taking change in cash transactions | 0.73 | 1.02 | 1.39 | 1.23 | 0.779 | 0.962 | 0.962 |
| 23. I am worried that I might catch the virus from handling money or using a debit machine | 0.87 | 1.06 | 1.15 | 0.56 | 0.739 | 0.962 | 0.962 |
| 24. I am worried that my mail has been contaminated by mail handlers | 0.65 | 1.01 | 1.58 | 1.87 | 0.730 | 0.962 | 0.963 |
| 25. I had trouble concentrating because I kept thinking about the virus | 0.75 | 1.01 | 1.32 | 1.21 | 0.713 | 0.962 | 0.963 |
| 26. Disturbing mental images about the virus popped into my mind against my will | 0.57 | 0.95 | 1.81 | 2.77 | 0.637 | 0.963 | 0.963 |
| 27. I had trouble sleeping because I worried about the virus | 0.55 | 0.94 | 1.68 | 2.05 | 0.711 | 0.962 | 0.963 |
| 28. I thought about the virus when I didn't mean to | 0.68 | 1.02 | 1.37 | 0.95 | 0.694 | 0.962 | 0.963 |
| 29. Reminders of the virus caused me to have physical reactions, such as sweating or a pounding heart | 0.46 | 0.87 | 1.97 | 3.33 | 0.689 | 0.963 | 0.963 |
| 30. I had bad dreams about the virus | 0.31 | 0.75 | 2.93 | 8.71 | 0.616 | 0.963 | 0.963 |
| 31. Searched the Internet for treatments for COVID-19 | 0.94 | 1.03 | 0.88 | -0.01 | 0.529 | 0.963 | 0.964 |
| 32. Asked health professionals (e.g., doctors or pharmacists) for advice about COVID-19 | 0.63 | 0.90 | 1.34 | 1.11 | 0.544 | 0.963 | 0.963 |
| 33. Checked YouTube videos about COVID-19 | 0.73 | 0.98 | 1.22 | 0.81 | 0.481 | 0.964 | 0.964 |
| 34. Checked your own body for signs of infection (e.g., taking your temperature) | 0.85 | 1.11 | 1.12 | 0.31 | 0.605 | 0.963 | 0.963 |
| 35. Sought reassurance from friends or family about COVID-19 | 0.48 | 0.88 | 2.08 | 4.14 | 0.616 | 0.963 | 0.963 |
| 36. Checked social media posts concerning COVID-19 | 1.11 | 1.19 | 0.81 | -0.34 | 0.481 | 0.964 | 0.964 |

Min—minimum (0 –not at all / never); Max—maximum (4 –extremely / almost always); SD -standard deviation; Skew—skewness; Kurt–kurtosis; CI–TC—Corrected Item-Total Correlation; Alpha—Cronbach's alpha coefficient; Omega—McDonald's omega coefficients.

**Table 2. Average values, Cronbach's alpha and correlation coefficients between domains/factors.**

| Domains factors) | Mean | SD | Alpha | Omega | DAN | SEC | XEN | CON | TSS | CHE |
|---|---|---|---|---|---|---|---|---|---|---|
| DAN | 9.69 | 5.56 | 0.890 | 0.890 | 1.000 | - | - | - | - | - |
| SEC | 4.57 | 5.65 | 0.939 | 0.939 | 0.553 | 1.000 | - | - | - | - |
| XEN | 7.51 | 6.51 | 0.944 | 0.945 | 0.459 | 0.516 | 1.000 | - | - | - |
| CON | 5.49 | 5.45 | 0.926 | 0.926 | 0.578 | 0.639 | 0.718 | 1.000 | - | - |
| TSS | 3.33 | 4.81 | 0.932 | 0.935 | 0.467 | 0.591 | 0.478 | 0.689 | 1.000 | - |
| CHE | 4.76 | 4.81 | 0.874 | 0.877 | 0.403 | 0.498 | 0.398 | 0.551 | 0.664 | 1.000 |

DAN—Danger subscale; SEC—Socio-economic consequences subscale; XEN—Xenophobia subscale; CON—Contamination subscale; TSS—Traumatic Stress subscale; CHE—Compulsive Checking subscale; SD—standard deviation; Alpha—Cronbach's alpha coefficient.

analysis. The Serbian CSS with 36 items and a six-factor structure showed a measurement model with a satisfactory fit for our population (CMIN/DF = 4.391; p = 0.001; GFI = 0.991; AGFI = 0.847; CFI = 0.996; NNFI = 0.995; RMSEA = 0.025). On the other hand, a five-factor structure was somewhat less adequate (CMIN/DF = 5.881; p = 0.001; GFI = 0.708; AGFI = 0.799; CFI = 0.829; NNFI = 0.815; RMSEA = 0.097). Therefore, we opted to accept and further explore the six-factor Serbian CSS.

The observed low covariance between the six factors confirmed adequate construct validity of the CSS (Fig 1). The only covariance above 0.5 was observed for CON and XEN domains. Yet, covariance was observed between items 30, 35, 29 and 30; 20 and 21; 10 and 12; 7, 10 and 8; and 3 and 4, indicating that they were part of a similar construct. Nevertheless, as the overall model was adequate, we opted to keep all the original items in the Serbian CSS.

## Criterion validity

Total PSS scores in our sample were quite high, with a mean of 21.4 (SD = 5.25) and ranged of 7 to 46. Only 60 participants had stress levels below the highest average of norm groups (14 points) measured for the United States (23).

The CSS total and all domain scores significantly positively correlated with PSS total scores. Moreover, inter-correlations for CSS domains were all statistically significant indicating good convergent validity (Table 3).

Based on the ROC curve analysis, the stress levels measured by the CSS adequately explained 65.9% of cases compared with PSS (p = 0.001). The cut-off level of CSS above which the stress could be considered as high in our study was 24.5 score (specificity = 62.8%; sensitivity = 61.7%; Fig 2).

## Discussion

Results of this study suggest that the CSS administered in Serbian language among Serbians has a 6-factorial structure, which mirrors the original construct of the questionnaire. There was no compelling metric evidence that any of the items should be omitted. The questionnaire showed adequate overall internal consistency, whereby all coefficients were >0.8, which demonstrates excellent internal consistency of the domains. The CSS is comparable to another well-established questionnaire examining stress, the PSS. Based on the ROC analysis, the CSS could potentially be used in rapid triage of persons who are at risk of higher stress levels requiring clinical attention.

Our study was conducted in a community sample of adults, as was done in the original validation study [19]. The psychometric properties of the CSS in Persian have also been examined in a sample of persons with anxiety and obsessive-compulsive disorders in Iran [20]. In both studies, the participants scored the highest on the DAN scale, which is consistent with our

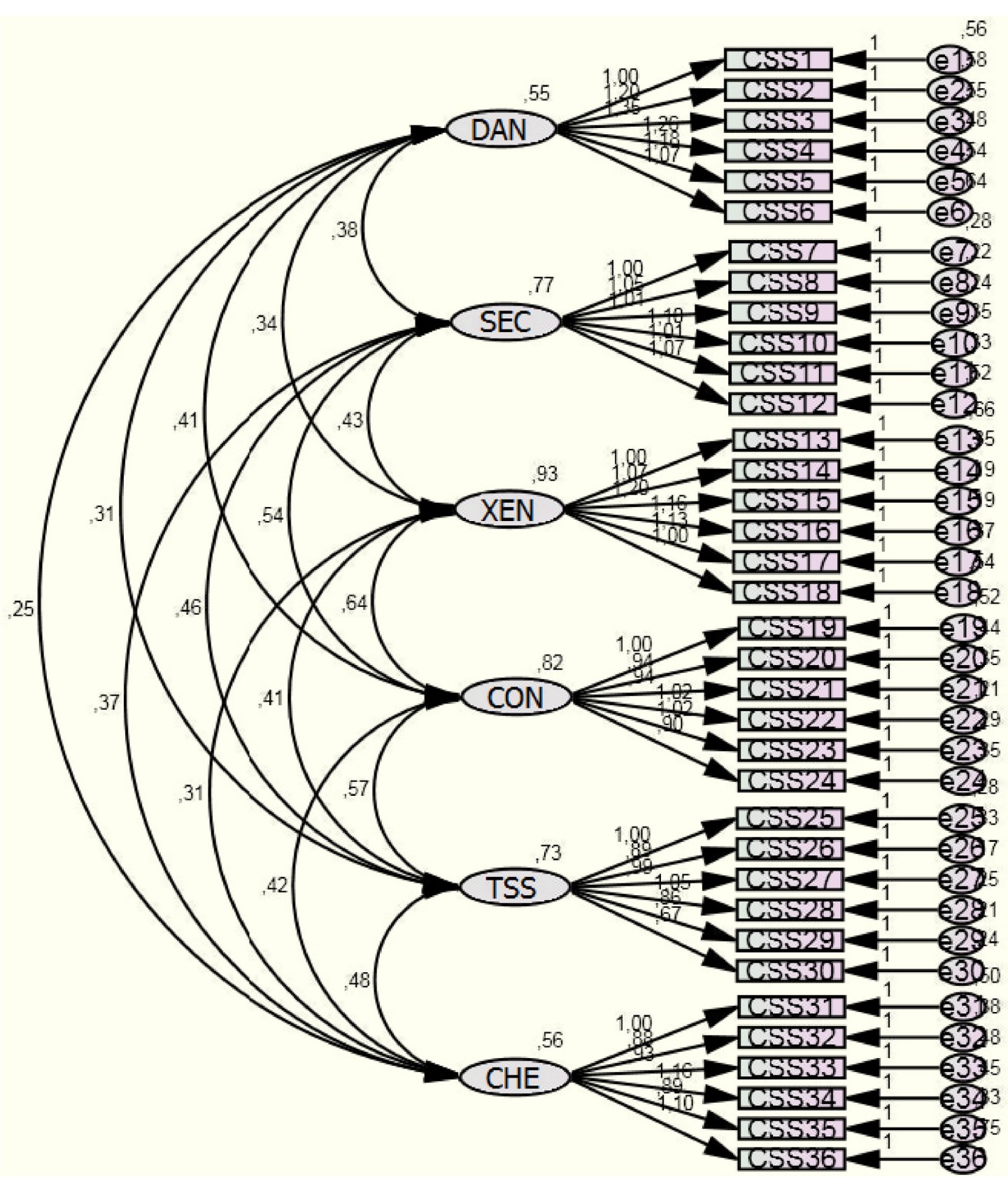

**Fig 1. Confirmatory factor analysis for COVID Stress Scales.**

**Table 3. Correlations between the COVID Stress Scales and Perceived Stress Scale total scores.**

| Domains | | DAN | SEC | XEN | CON | TSS | CHE | Total CSS |
|---|---|---|---|---|---|---|---|---|
| PSS total score | Rho | 0.291 | 0.264 | 0.209 | 0.197 | 0.323 | 0.218 | 0.317 |
| | p | 0.001 | 0.001 | 0.001 | 0.001 | 0.001 | 0.001 | 0.001 |
| DAN | Rho | 1.000 | 0.537 | 0.444 | 0.545 | 0.413 | 0.366 | 0.732 |
| | p | - | 0.001 | 0.001 | 0.001 | 0.001 | 0.001 | 0.001 |
| SEC | Rho | - | 1.000 | 0.524 | 0.602 | 0.511 | 0.416 | 0.772 |
| | p | - | - | 0.001 | 0.001 | 0.001 | 0.001 | 0.001 |
| XEN | Rho | - | - | 1.000 | 0.723 | 0.431 | 0.378 | 0.795 |
| | p | - | - | - | 0.001 | 0.001 | 0.001 | 0.001 |
| CON | Rho | - | - | - | 1.000 | 0.586 | 0.489 | 0.854 |
| | p | - | - | - | - | 0.001 | 0.001 | 0.001 |
| TSS | Rho | - | - | - | - | 1.000 | 0.577 | 0.704 |
| | p | - | - | - | - | - | 0.001 | 0.001 |
| CHE | Rho | - | - | - | - | - | 1.000 | 0.652 |
| | p | - | - | - | - | - | - | 0.001 |
| Total CSS score | Rho | - | - | - | - | - | - | - |
| | p | - | - | - | - | - | - | - |

DAN—Danger subscale; SEC—Socio-economic consequences subscale; XEN—Xenophobia subscale; CON—Contamination subscale; TSS—Traumatic Stress subscale; CHE—Compulsive Checking subscale; PSS–Perceived stress scale; CSS–COVID Stress Scales.

results [19, 20]. On the other hand, our participants scored the lowest on the TSS scale, while in Iran, for example, the lowest scores were observed on the SEC scale [20]. Yet, the selection of the examined population in Iran was considerably different from ours (i.e., patients from psychiatric hospitals and clinical centers vs. community sample) and could have, therefore, contributed to the observed differences.

The study sample was representative of the population in both countries as there were no significant differences in socio-demographic data of questionnaire respondents and the general populations of Republic of Serbia and Republic of Srpska. The mean age of our study participants was 38.2 +/- 14.1 years. In Serbia children up to 14 years encompass 14.3%, 65.0% of citizens are aged 15–65 years, while seniors over 65 years make 20.7% of the population. In Republic of Srpska the age group up to 14 years includes 14.1% of the population, 68.8% is aged 15–65, while 17.1% of the population is aged over 65 years [30].

In our study 52.8% participants were males and 47.2% females. Out of 6,945,235 inhabitants of the Republic of Serbia 48.7% are males and 51.3% are females, while out of 1,147,902 people who live in the Republic of Srpska 48.9% are males and 51.1% are females. In Serbia 73.1% of the population is married, while 57.3% of the population of Republic of Srpska is married. In our study the number of married participants was somewhat lower, but being married was still the most common relationship status (45.3%) [31].

The majority of our study participants reported having secondary education (54%) and being permanently employed (50.8%). About 2.9 million people in Serbia are employed and 292,000 are unemployed. As for Republic of Srpska about 350,670 people are employed and 118,189 are unemployed. In Serbia about 13.7% of the population has no or incomplete primary education, 20.8% have completed only primary school, 48.9% have completed secondary school, and 16.3% of people have higher or high i.e. university level of education. In Republic of Srpska about 15.2% of the population has no or incomplete primary education, 21.2% have completed only primary school, 50.6% have completed secondary school, and 12.8% of people have higher or high i.e. university level of education.

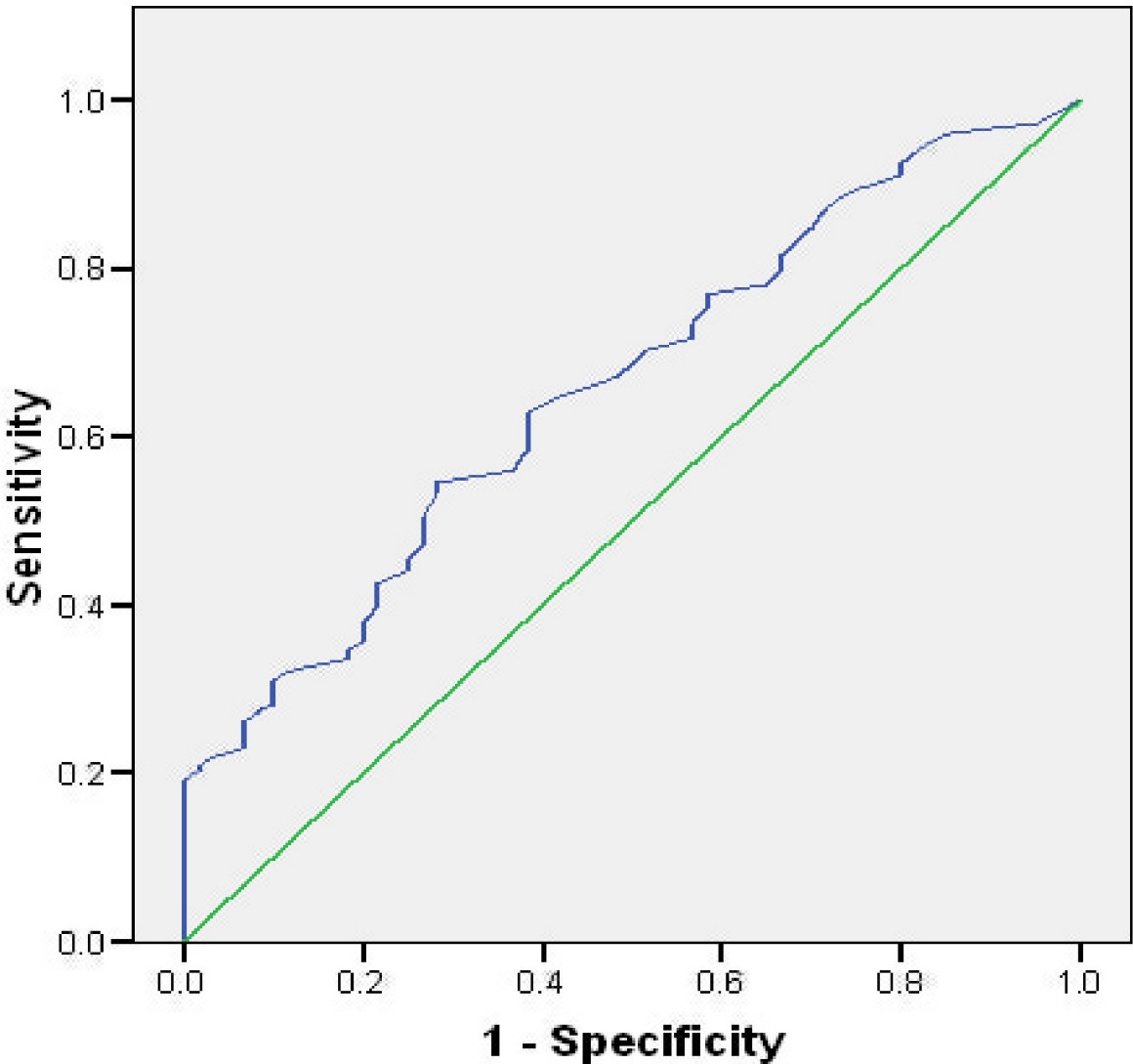

**Fig 2. ROC analysis of COVID Stress Scales according to Perceived Stress Scale.**

Results of our study could be additionally explained by the timing of our survey relative to the onset of the pandemic. Specifically, the index case in Serbia was identified in the first week of March 2020; yet, it is possible that the SARS-CoV2 was circulating in the population in early 2020 or late 2019 [32]. The current study was conducted during September to December of 2020, which may have allowed enough time for individuals to adjust to the circumstances of the pandemic, such as social distancing and working remotely, so that the experience was not perceived as traumatic at this point in time. If we had surveyed the population at the very beginning of the pandemic it is possible that the results could have been different. Nevertheless, the pandemic is still ongoing in Serbia. While the frequency of newly diagnosed people with COVID-19 increased in June and July and decreased in August and September, the highest incidence rates per 100,000 were observed in November and early December 2020. Consequently, the study period covered both low and high rates of COVID-19 enabling us to test their potentially different impact on stress level in our population.

Differences in the factor-structure of the CSS may reflect a specific context or purpose of the questionnaire. We found that 6-factor solution is suited for our community sample. Contrary, the original validation study, utilizing parallel analyses, demonstrated that a 5-factor solution was sufficiently stable for the Canadian and US community-based samples [19]. This solution was replicated in Iran among persons with anxiety and obsessive-compulsive disorders [20]. In this 5-factor model, DAN and CON subscales were merged and observed as a single construct [19, 20]. Our findings suggest that in the Serbian population, a distinction between perceptions of the pandemic as dangerous and disrupting everyday functioning on the one hand, and getting exposed to virus in the immediate environment on the other hand, should be made.

We observed that the CSS internal consistency as measured by both the alpha and omega coefficients was satisfactory. The original validation study used only the alpha coefficient to test the internal consistency of the scale. The validation of the CSS in the Iranian sample used both alpha and omega coefficients, which were similar to ours ($\geq 0.88$). It has been discussed that the omega coefficient is more reliable than the alpha coefficient as it accounts for the variability of the covariance [33]. For this reason, it is recommended that the omega coefficient is included in psychometric testing.

Previous validation studies observed good convergent validity of the CSS in English and Persian [19, 20]. We administered the PSS alongside the CSS to examine whether the Serbian CSS could distinguish the persons who experience more intense stress from those who do not. We observed that the level of sensitivity and specificity of the CSS in Serbian might be utilized as triage orientation to identify potential persons who need systematic mental health support. For this reason, further exploration of CSS in screening for COVID-19-related distress is warranted.

## Strengths and limitations

The strength of this study is the fact that we selected a community-based sample (i.e., general population from several cities, including the capital of the Republic of Serbia) in two countries. Having the adequate participant-to-questionnaire item ratio the conditions for factor analysis were satisfied and bias arising from the number of observations was minimized [23]. Finally, the inclusion of the McDonald's omega coefficient strengthened the interpretation of the scale internal consistency and reliability.

Some limitations need to be mentioned. The study sample might not have equal probability of selection, but is to an extent biased in favor of residents with reasons to go to municipality headquarters. At the time of pandemics a significant number of people might have been reluctant to go out if not absolutely necessary, as suggested by the medical authorities. Therefore, these individuals have not been included in our study. However, in Serbia and Bosnia and Herzegovina, many people visit municipal government offices to complete with different administrative procedures regardless of their gender, age, socio-economic status or political preferences. Moreover, we did not include a longitudinal follow-up of the study sample. As such, we were unable to re-test the selected population. In this way, the examination of stability of the CSS in Serbian language is missing. We did not include testing of any other parameter except the PSS, so this study is limited in terms of criterion validity. The inclusion of depression might have been useful to strengthen the convergent validity.

## Conclusion

In conclusion, this study suggests that the CSS in the Serbian general population is valid and reliable questionnaire to assess distress related to the COVID-19 pandemic. The CSS in

Serbian language incorporates six domains. We recommend this questionnaire in further research about mental health during the current pandemic as well as in research on the mental health impacts of future epidemics and pandemics.

## Supporting information

**S1 File. Additional description of the COVID Stress Scales translation and modification and adjustment of COVID Stress Scales to Serbian language.**
(DOC)

**S1 Data.**
(XLS)

## Acknowledgments

We are grateful to Ms. Bojana Milic, Bachelor of Laws, for help with administration and data entry. We are thankful to all the municipality administrations in Serbia and Republic of Srpska (Bosnia and Herzegovina) and all study participants who made this study possible.

## Author Contributions

**Conceptualization:** Marija Milic, Jelena Dotlic, Geoffrey S. Rachor, Gordon J. G. Asmundson, Bojan Joksimovic, Jasmina Stevanovic, Dragoslav Lazic, Zorica Stanojevic Ristic, Jelena Subaric Filimonovic, Nikoleta Radenkovic, Milica Cakic, Tatjana Gazibara.

**Data curation:** Marija Milic, Jelena Dotlic, Jasmina Stevanovic, Dragoslav Lazic, Jelena Subaric Filimonovic, Nikoleta Radenkovic, Milica Cakic, Tatjana Gazibara.

**Formal analysis:** Marija Milic, Jelena Dotlic, Geoffrey S. Rachor, Gordon J. G. Asmundson, Bojan Joksimovic, Zorica Stanojevic Ristic, Tatjana Gazibara.

**Investigation:** Marija Milic, Jelena Dotlic, Geoffrey S. Rachor, Gordon J. G. Asmundson, Bojan Joksimovic, Jasmina Stevanovic, Dragoslav Lazic, Zorica Stanojevic Ristic, Jelena Subaric Filimonovic, Nikoleta Radenkovic, Milica Cakic, Tatjana Gazibara.

**Methodology:** Marija Milic, Jelena Dotlic, Geoffrey S. Rachor, Gordon J. G. Asmundson, Bojan Joksimovic, Jasmina Stevanovic, Dragoslav Lazic, Zorica Stanojevic Ristic, Jelena Subaric Filimonovic, Nikoleta Radenkovic, Milica Cakic, Tatjana Gazibara.

**Project administration:** Tatjana Gazibara.

**Supervision:** Tatjana Gazibara.

**Validation:** Geoffrey S. Rachor, Gordon J. G. Asmundson.

**Writing – original draft:** Marija Milic, Jelena Dotlic, Tatjana Gazibara.

**Writing – review & editing:** Geoffrey S. Rachor, Gordon J. G. Asmundson, Bojan Joksimovic, Jasmina Stevanovic, Dragoslav Lazic, Zorica Stanojevic Ristic, Jelena Subaric Filimonovic, Nikoleta Radenkovic, Milica Cakic.

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
