## [Decision Letter · Decision Letter 0]

21 Jun 2021

PONE-D-21-14941

Validity and reliability of the Serbian COVID Stress Scales

PLOS ONE

Dear Authors,

Thank you for submitting your manuscript to PLOS ONE. After careful consideration, we feel that it has merit but does not fully meet PLOS ONE’s publication criteria as it currently stands. Therefore, we invite you to submit a revised version of the manuscript that addresses the points raised during the review process.

We look forward to receiving your revised manuscript.

Kind regards,

Marcel Pikhart

Academic Editor

PLOS ONE

Journal Requirements:

Reviewers' comments:

Reviewer's Responses to Questions

**Comments to the Author**

1. Is the manuscript technically sound, and do the data support the conclusions?

Reviewer #1: Partly

Reviewer #2: Yes

Reviewer #3: Yes

Reviewer #4: Partly

2. Has the statistical analysis been performed appropriately and rigorously? 

Reviewer #1: Yes

Reviewer #2: Yes

Reviewer #3: Yes

Reviewer #4: Yes

3. Have the authors made all data underlying the findings in their manuscript fully available?

Reviewer #1: Yes

Reviewer #2: Yes

Reviewer #3: Yes

Reviewer #4: No

4. Is the manuscript presented in an intelligible fashion and written in standard English?

Reviewer #1: Yes

Reviewer #2: Yes

Reviewer #3: Yes

Reviewer #4: Yes

5. Review Comments to the Author

Reviewer #1: This paper constructs and then validates a Serbian language version of the CSS using standard statistical techniques. The validation appears to be competently done and in order.

My criticism of the paper is that it fails to account for a likely bias arising from the method of respondent recruitment. This is done by approaching individuals accessing municipal government offices, suggesting that this may select for individuals with higher levels of political and/or economic resources relative to the general population. The degree of this bias could be assessed by comparing the sociodemographic profile of the respondents to that of the country as a whole, and to the municipalities included in the study. As it now stands the only mention of demographic composition is the sex ratio of the respondents, but not the sex ratio in Serbian countries.

Reviewer #2: Overall this is a good piece of research. I have only a few comments that should be easy to act upon. Line 62, "every day" should be "everyday" ( 1 word, not 2). There seem to be differences in the meaning of "municipality" (lines 94-96) between Serbia and the U.S. (where I am located). Here cities are a kind of municipality and other municipalities are contiguous to them. (Generally, counties here are divided into municipalities, so there are also many rural municipalities, in addition to suburban, but the city, proper, is always its own municipality.) The text suggests that sets of municipalities are contained within cities. Clarification on the organization of relevant political geography and comparison with the EU and U.S. systems might be appropriate. Lines 96 to 98 describe a convenience sample, which means that a caveat or study limitation is that the sample does not have equal probability of selection (EPSEM), but is biased in favor of residents with reasons to go to municipality headquarters. (Those with the greatest reluctance to go out in the pandemic may have similar characteristics among themselves that correlate with the variables measured by the CSS.) Line 101, "signing informed consent" is implied and may be deleted. Line 103, not fulfilling the inclusion criteria is redundant and may thus be deleted. The description of the translation process (lines 140 - 162, but especially lines 140 - 146) is a most welcome detail; although lines 147-162 seem to be a bit of a digression. I think perhaps lines 147-162 plus Table 1 could be lifted out of the main text and placed together in an appendix. Line 185, it might be helpful for some readers to have the synonym of "composite reliability" (i.e., "construct reliability") noted. Line 233 contains an English grammatical error: "skewness" is singular whereas "kurtoses" is plural--unless there really is only 1 skewness together with multiple kurtoses, then both here should be plural. (The plural of skewness is "skewnesses"). Line 314, I think should perhaps not be referring to the "CSS in Serbian language" but the "CSS among Serbians, administered in Serbian" (yes, it's a fine point, and most readers will understand it correctly, but if the sample were of Serbians living abroad in, say, Chicago, the 5-factorial structure might apply instead of the 6-factorial structure). Line 376 "did not include the longitudinal follow-up" implies that a follow up was planned but not carried out. (Also, "follow-up" should not be hyphenated because it is not used as a two-word modifier, i.e., as an adjective or adverb.) If you say instead that you did not include "a longitudinal follow up of the study sample" you avoid unnecessarily creating the impression that you failed to do something that you had planned.

Reviewer #3: Data has been provided to support the conclusions as authors state within the Supporting Information files.

Statistical analysis has been performed according Cronbach's alpha and McDonald’s omega coefficients. TAs stated by the authors, the study was approved by the Ethics Committee of the University of Pristina temporarily settled in Kosovska Mitrovica (Approval no. 10-1285/1), informed consent being signed by all participants.

According the Data availability Statement, all data are available without restrictions.

The manuscript is presented in an intelligible fashion and written in standard English.

Reviewer #4: Dear Editor,

Thanks for the opportunity to review the manuscript titled, "Validity and reliability of the Serbian COVID Stress Scales" by Milic et al. This study translated and validated COVID Stress Scales in Serbian, which is important and relevant to the study population. Overall, the manuscript is well-written. The authors used proper statistical tests to assess the psychometric properties of the translated scale. However, there are some issues that the authors need to address before the manuscript can be considered for publication. The following are my comments describing these issues.

1. The introduction lacks local perspective of the pandemic. How has the COVID-19 pandemic affected local people? What public health policy has been implemented in Republic of Serbia and Republic of Srpska? During or before the data collection period, any important event happened there? (Stricter preventive measures? Border control? Lockdown?) Please provide more background information for the study population.

2. (L. 63-65) Please elaborate this paragraph by providing more evidence on how these preventive measures or other aspects of the pandemic affect people’s mental health.

3. (L. 92-94) How was the number of cities selected in each country determined? Proportionate to population size? Logistic convenience?

4. (L. 101) Signing informed consent is not a proper inclusion criterion. Inclusion criteria should describe the property of your study population to whom your results are generalizable to.

5. (L. 102) Need to clarify what psychiatric disorders were considered “could evidently negatively impact self-perception and comprehension”.

6. (L. 103) “Not fulfilling the inclusion criteria” is not a proper exclusion criterion. Exclusion criteria should describe within your sample who were included based on your inclusion criteria, who should be excluded.

7. (L. 104) “In total, we approached 1,347 persons out of which approximately 71% fulfilled the study criteria.” Please provide number of people being excluded for each reason separately. How many people were excluded due to reporting psychiatric disorders? How many people were excluded due to providing less than 90% of answers?

8. (L. 112-114) How was the questionnaire administered? Self-administered? Research staff asked questions and participants answered? Paper and pen or electronic questionnaire?

9. (L. 136-137) Any cut-off available for the original PSS and adapted PSS in Serbian?

10. (L. 226) “In our sample, 25% (n=240) confirmed having contact with a COVID-19 positive person” Is there a question for family member/close friends tested positive for COVID-19?

11. (L. 227-228) “…but only 18.4% (n=177) respondents were tested for COVID-19 due to typical symptoms.” What were their test results? Any positive case?

12. Please add item numbers in Table 2.

13. (L. 257) I suggest using “Internal consistency” instead of “Reliability” here and in the discussion for accuracy as test-retest reliability was not established in this study.

14. (L. 280-281) To my understanding, discriminant validity refers to the measures of different constructs that theoretically should not be highly related to each other are not found to be highly correlated to each other. The authors did not compare CSS with another construct. Instead, only different domains within CSS were assessed, which cannot “confirm adequate discriminant validity of the CSS”.

15. Were there any missing data? How were the missing data handled?

16. (L. 307) Which method was used to identify the cut-off? The closest to (0,1) criteria (ER)? Youden Index? Concordance Probability Method?

17. (L. 370) Proper steps of sample size calculation should be presented in the methods section instead of the discussion section.

18. (L. 373) Please explain what “parallel analysis” refers to and how it “helps to minimize selection bias”.

6. PLOS authors have the option to publish the peer review history of their article (what does this mean?). If published, this will include your full peer review and any attached files.

Reviewer #1: No

Reviewer #2: No

Reviewer #3: **Yes: **Vladia Ionescu

Reviewer #4: No

---

## [Author Response · Author response to Decision Letter 0]

30 Sep 2021

Reviewer #1: 

My criticism of the paper is that it fails to account for a likely bias arising from the method of respondent recruitment. This is done by approaching individuals accessing municipal government offices, suggesting that this may select for individuals with higher levels of political and/or economic resources relative to the general population. The degree of this bias could be assessed by comparing the socio-demographic profile of the respondents to that of the country as a whole, and to the municipalities included in the study. As it now stands the only mention of demographic composition is the sex ratio of the respondents, but not the sex ratio in Serbian countries.

ANSWER 1: We are compelled to clarify the concept of municipality headquarters and rationale behind the selection of study participants who were doing some administrative business at their municipality. 

In Serbia and Republic of Srpska (Bosnia and Herzegovina), civil administration affairs are just starting to function online. The vast majority of people who require any administration related to taxation or are in need of official certificates (birth, death, marriage, divorce, etc) are required to come in person to the municipality office. Thus, many people visit their municipality office quite often. This does not depend on their political or economic resources i.e. all people regardless of their political stance and/or economic resources need to visit their municipality office at some point in time. Contrary, people with lower economic resources, from smaller towns as well as the older people mostly choose to visit the offices in person to do all sorts of administrative procedures. However, even young adults who are digitally literate visit the municipality offices whenever they need help with more complex procedures. Moreover, municipal government offices are not the center for political activities, but have solely administrative purpose. This very characteristic enabled us to sample people at random and include individuals of different socio-demographic backgrounds. 

We have now added this clarification in Methods (Subsection Selection of study participants). We have also added the short comparison of the socio-demographic profile of the respondents to that of the country as a whole in the Discussion section. 

Reviewer #2: 

Line 62, "every day" should be "everyday" (1 word, not 2).

ANSWER 1: Thank you for this correction. We have now changed “every day" to "everyday". 

There seem to be differences in the meaning of "municipality" (lines 94-96) between Serbia and the U.S. (where I am located). Here cities are a kind of municipality and other municipalities are contiguous to them. (Generally, counties here are divided into municipalities, so there are also many rural municipalities, in addition to suburban, but the city, proper, is always its own municipality.) The text suggests that sets of municipalities are contained within cities. Clarification on the organization of relevant political geography and comparison with the EU and U.S. systems might be appropriate. 

ANSWER 2: Indeed, differences between countries exist. We have now explained in more details how are the cities in Serbia organized. Moreover, we have now clarified that we selected the study participants inside municipality offices. 

In Republic of Serbia there are 5 regions (Belgrade, Vojvodina Region, Sumadija and Western Serbia region, Southern and Eastern Serbia region and Kosovo and Metohija region) with 197 municipalities (22). Rural and suburban regions are divided and clustered into municipalities according to population size and geographical localizations. On the other hand, towns and cities form municipalities based on the number of residents. Smaller towns and cities are considered as one municipality while larger cities are divided into different municipalities for easier administration. Therefore, in Serbia there are 17 municipalities in the Belgrade region, 45 in the region of Vojvodina, 53 in the region of Sumadija and Western Serbia, 53 in the region of Southern and Eastern Serbia 29 in the region of Kosovo and Metohija (22). However, because some cities had just one and others multiple municipalities, to overcome the discrepancies in the population size, we decided to distribute the questionnaires in only one randomly chosen municipality per city. To enable random selection of people and include individuals of different socio-demographic backgrounds we decided to approach and invite to participate in the study all people who came to the chosen municipality office headquarters to engage in regular administrative business. 

Lines 96 to 98 describe a convenience sample, which means that a caveat or study limitation is that the sample does not have equal probability of selection (EPSEM), but is biased in favor of residents with reasons to go to municipality headquarters. (Those with the greatest reluctance to go out in the pandemic may have similar characteristics among themselves that correlate with the variables measured by the CSS.) 

ANSWER 3: In Serbia and Republic of Srpska (Bosnia and Herzegovina), civil administration affairs are just starting to function online. The vast majority of people who require any administration related to taxation or are in need of official certificates (birth, death, marriage, divorce, etc) are required to come in person to the municipality office. Thus, many people visit their municipality office quite often. This does not depend on their political or economic resources i.e. all people regardless of their political stance and/or economic resources need to visit their municipality office at some point in time. Contrary, people with lower economic resources (no internet access), from smaller towns as well as the older people mostly choose to visit the offices in person to do all sorts of administrative procedures. However, even young adults who are digitally literate visit the municipality offices whenever they need help with more complex procedures. Moreover, municipal government offices are not the center for political activities, but have solely administrative purpose. This very characteristic enabled us to sample people at random and include individuals of different socio-demographic backgrounds. 

We have now added this clarification in Methods (Subsection Selection of study participants). We have also added the short comparison of the socio-demographic profile of the respondents to that of the country as a whole in the Discussion section. 

Nevertheless, we do agree that some bias might come from the fact that in time of pandemic people with the least reasons to go to municipality headquarters might have stayed in their homes and consequently have not been included in our study. We have now addressed this issue in the limitations section of our manuscript. 

Line 101, "signing informed consent" is implied and may be deleted. 

ANSWER 4: We have now deleted this phrase. 

Line 103, not fulfilling the inclusion criteria is redundant and may thus be deleted. 

ANSWER 5: We have now deleted this phrase. 

The description of the translation process (lines 140 - 162, but especially lines 140 - 146) is a most welcome detail; although lines 147-162 seem to be a bit of a digression. I think perhaps lines 147-162 plus Table 1 could be lifted out of the main text and placed together in an appendix. 

ANSWER 6: We have now placed the text about the translation process and the Supplemental Table 1 into Appendix as suggested. 

Line 185, it might be helpful for some readers to have the synonym of "composite reliability" (i.e., "construct reliability") noted. 

ANSWER 7: We have now written both synonyms in the text. 

Line 233 contains an English grammatical error: "skewness" is singular whereas "kurtoses" is plural--unless there really is only 1 skewness together with multiple kurtoses, then both here should be plural. (The plural of skewness is "skewnesses"). 

ANSWER 8: We have now corrected this grammatical error and written both words in singular. 

Line 314, I think should perhaps not be referring to the "CSS in Serbian language" but the "CSS among Serbians, administered in Serbian" (yes, it's a fine point, and most readers will understand it correctly, but if the sample were of Serbians living abroad in, say, Chicago, the 5-factorial structure might apply instead of the 6-factorial structure). 

ANSWER 9: We agree, and we have now rewritten this sentence as suggested. 

Line 376 "did not include the longitudinal follow-up" implies that a follow up was planned but not carried out. (Also, "follow-up" should not be hyphenated because it is not used as a two-word modifier, i.e., as an adjective or adverb.) If you say instead that you did not include "a longitudinal follow up of the study sample" you avoid unnecessarily creating the impression that you failed to do something that you had planned.

ANSWER 10: Thank you for this very useful clarification. We have now rewritten the sentence as suggested. 

Reviewer #3: 

Data has been provided to support the conclusions as authors state within the Supporting Information files.

Statistical analysis has been performed according Cronbach's alpha and McDonald’s omega coefficients. TAs stated by the authors, the study was approved by the Ethics Committee of the University of Pristina temporarily settled in Kosovska Mitrovica (Approval no. 10-1285/1), informed consent being signed by all participants.

According the Data availability Statement, all data are available without restrictions.

The manuscript is presented in an intelligible fashion and written in standard English.

ANSWER 1: Thank you for your comments. 

Reviewer #4: 

1. The introduction lacks local perspective of the pandemic. How has the COVID-19 pandemic affected local people? What public health policy has been implemented in Republic of Serbia and Republic of Srpska? During or before the data collection period, any important event happened there? (Stricter preventive measures? Border control? Lockdown?) Please provide more background information for the study population.

ANSWER 1: We agree. We have now explained the perspective of the pandemic in Serbia in the Introduction section. 

Introduction, 1st paragraph:

"The first case of COVID-19 in the Republic of Serbia was identified on March 6th, 2020. Since then, to the end of June 2021, more than 700,000 confirmed cases of COVID-19 were registered, and more than 7,000 people died due to COVID-19-related complications. The dynamic of the epidemic over the past 15 months in Serbia could be divided in distinctive three waves (March to end-of-May 2020; July to the end of August 2020; October 2020 to the mid-May 2021, when the largest number of people caught the novel coronavirus and died as a result) (1)."

Introduction, 3rd paragraph:

"Over the first wave of the pandemic, both Republic of Serbia and Republic of Srpska (Bosnia and Herzegovina) entered a lockdown, although residents who were living abroad were allowed to enter the country. During lockdown, people aged 65 and above were not allowed to leave their homes and were allocated a time slot once a week to shop for groceries, while curfew for all citizens lasted throughout the entire week-ends. Reopening was gradual along with the improvement of the epidemiological situation (17). Over the following months and other epidemic waves, various preventive and control measures were in place (working/schooling from home, closure of public spaces and ban of celebrations and gatherings of more than 5 people, reduction of working hours of cafes and restaurants, mandatory use of face masks inside and outside, social distancing, etc) except lockdown. The borders remained open during second and third epidemic wave although incoming people were requested to present certificate of negative PCR coronavirus test (17). In Serbia, a national psychological telephone support service during lockdown. One quarter of people who used this service was older than 70 years, 63.6% were women, and one third reported having symptoms of anxiety and feelings of tension during the COVID-19 epidemic (18)." 

2. (L. 63-65) Please elaborate this paragraph by providing more evidence on how these preventive measures or other aspects of the pandemic affect people’s mental health.

ANSWER 2: We have now added a more detailed explanation how preventive measures or other aspects of the pandemic affect people’s mental health. 

Introduction, 3rd paragraph:

"Over the first wave of the pandemic, both Republic of Serbia and Republic of Srpska (Bosnia and Herzegovina) entered a lockdown, although residents who were living abroad were allowed to enter the country. During lockdown, people aged 65 and above were not allowed to leave their homes and were allocated a time slot once a week to shop for groceries, while curfew for all citizens lasted throughout the entire weekends. Reopening was gradual along with the improvement of the epidemiological situation (17). Over the following months and the epidemic waves, various preventive and control measures were in place (working/schooling from home, closure of public spaces and ban of celebrations and gatherings of more than 5 people, reduction of working hours of cafes and restaurants, mandatory use of face masks inside and outside, social distancing, etc) except lockdown. The borders remained open during second and third epidemic wave although incoming people were requested to present certificate of negative PCR coronavirus test (17). In Serbia, a national psychological telephone support service during lockdown. One quarter of people who used this service was older than 70 years, 63.6% were women, and one third reported having symptoms of anxiety and feelings of tension during the COVID-19 epidemic (18)." 

3. (L. 92-94) How was the number of cities selected in each country determined? Proportionate to population size? Logistic convenience?

ANSWER 3: Now we explained the process of random selection of cities selected in each country. 

Methods, Setting, 1st paragraph: 

"We printed the names of all 29 official administrative cities in Serbia and 8 in Republic of Srpska on separate slips of paper, folded them for blinding and placed them in a in a non-transparent container. In table of random numbers, we hit number nine and consequently draw from the container every ninth slip of paper with city names. In Republic of Serbia there are 5 regions (Belgrade, Vojvodina Region, Sumadija and Western Serbia region, Southern and Eastern Serbia region and Kosovo and Metohija region) with 197 municipalities (22). Rural and suburban regions are divided and clustered into municipalities according to population size and geographical localizations. On the other hand, towns and cities form municipalities based on the number of residents. Smaller towns and cities are considered as one municipality while larger cities are divided into different municipalities for easier administration. Therefore, in Serbia there are 17 municipalities in the Belgrade region, 45 in the region of Vojvodina, 53 in the region of Sumadija and Western Serbia, 53 in the region of Southern and Eastern Serbia 29 in the region of Kosovo and Metohija (22). However, because some cities had just one and others multiple municipalities, to overcome the discrepancies in the population size, we decided to distribute the questionnaires in only one randomly chosen municipality per city. We printed the names of all municipalities on separate slips of paper, folded them for blinding and placed them in one non-transparent container per city after which we chose one municipality from every container." 

4. (L. 101) Signing informed consent is not a proper inclusion criterion. Inclusion criteria should describe the property of your study population to whom your results are generalizable to.

ANSWER 4: We have now deleted this criterion. 

5. (L. 102) Need to clarify what psychiatric disorders were considered “could evidently negatively impact self-perception and comprehension”.

ANSWER 5: We have now explained in more details that we considered that all psychiatric disorders classified with the F diagnosis according to WHO ICD-10 were considered to evidently negatively impact self-perception and comprehension. Moreover, we explained better that we excluded those persons who had mental disorders clearly diagnosed by their doctors. 

F0: Organic, including symptomatic, mental disorders

F1: Mental and behavioral disorders due to use of psychoactive substances

F2: Schizophrenia, schizotypal and delusional disorders

F3: Mood [affective] disorders

F4: Neurotic, stress-related and somatoform disorders

F5: Behavioral syndromes associated with physiological disturbances and physical factors

F6: Disorders of personality and behavior in adult persons

F7: Mental retardation

F8: Disorders of psychological development

F9: Behavioral and emotional disorders with onset usually in childhood and adolescence

Methods, Selection of participants, 2nd paragraph:

"The exclusion criteria were: reporting psychiatric disorders previously diagnosed by a physician that could evidently negatively impact self-perception and comprehension (all F-coded diagnoses according to the International Classification of Diseases, 10th revision)"

6. (L. 103) “Not fulfilling the inclusion criteria” is not a proper exclusion criterion. Exclusion criteria should describe within your sample who were included based on your inclusion criteria, who should be excluded.

ANSWER 6: We have now deleted this criterion. 

7. (L. 104) “In total, we approached 1,347 persons out of which approximately 71% fulfilled the study criteria.” Please provide number of people being excluded for each reason separately. How many people were excluded due to reporting psychiatric disorders? How many people were excluded due to providing less than 90% of answers?

ANSWER 7: We have now presented all the data regarding our sampling methodology. 

Methods, Selection of study participants, 2nd paragraph:

"We excluded 19 individuals due to confirmed psychiatric and 367 due to not fulfilling 90% of the questionnaire."

8. (L. 112-114) How was the questionnaire administered? Self-administered? Research staff asked questions and participants answered? Paper and pen or electronic questionnaire?

ANSWER 8: We have now explained that the questionnaire was self-administered in paper form. 

Methods, Instruments, 1st paragraph:

"All instruments applied in the study were self-administered in paper-and-pencil form."

9. (L. 136-137) Any cut-off available for the original PSS and adapted PSS in Serbian?

ANSWER 9: In the available literature norms for the original PSS are based on US population and encompass the mean +/- sd values of PSS total score according to gender, age groups and race (Cohen S, Kamarck T, Mermelstein R. A global measure of perceived stress. J Health Soc Behav. 1983;24(4):385-96). However, there currently there are no PSS norms or cut-off values available for Serbian population. Therefore, we were compelled to compare findings of our population with the norms for the US citizens. 

10. (L. 226) “In our sample, 25% (n=240) confirmed having contact with a COVID-19 positive person” Is there a question for family member/close friends tested positive for COVID-19?

ANSWER 10: Yes, the general questionnaire encompasses the question regarding COVID-19 positive family members / close friends / colleagues / neighbors or other acquaintances. However, we opt to explore this in more detail in one of the forthcoming analyses.

11. (L. 227-228) “…but only 18.4% (n=177) respondents were tested for COVID-19 due to typical symptoms.” What were their test results? Any positive case?

ANSWER 11: Yes, we had 52 (3.9%) of respondents who had COVID 19 at some point. We now added this explanation to the text but only in short as we will explore these finding in more details in future. 

Results, Description of the study sample, 1st paragraph:

"In our sample, 25% (n=240) confirmed having contact with a COVID-19 positive person, but only 18.4% (n=177) respondents were tested for COVID-19 due to typical symptoms, out of which 52 (3.9%) had COVID 19 at some point."

12. Please add item numbers in Table 2.

ANSWER 12: We have added the item numbers in the Table. 

13. (L. 257) I suggest using “Internal consistency” instead of “Reliability” here and in the discussion for accuracy as test-retest reliability was not established in this study.

ANSWER 13: We changed reliability to internal consistency throughout the text where indicated. 

14. (L. 280-281) To my understanding, discriminant validity refers to the measures of different constructs that theoretically should not be highly related to each other are not found to be highly correlated to each other. The authors did not compare CSS with another construct. Instead, only different domains within CSS were assessed, which cannot “confirm adequate discriminant validity of the CSS”.

ANSWER 14: We have now corrected this error and rewritten this sentence to indicate that we only assessed construct validity. 

Results, Construct validity, 1st paragraph:

"The observed low covariance between the six factors confirmed adequate construct validity of the CSS."

15. Were there any missing data? How were the missing data handled?

ANSWER 15: Missing data were non-existent. Participants who did not fulfill the 90% of the questionnaire were excluded, which enabled us to have an entirely complete database. 

16. (L. 307) Which method was used to identify the cut-off? The closest to (0,1) criteria (ER)? Youden Index? Concordance Probability Method?

ANSWER 16: The cut-off level of CSS was identified using the ROC analysis. The cut-off level was the one with the best proportion (similar values) of sensitivity and specificity.

17. (L. 370) Proper steps of sample size calculation should be presented in the methods section instead of the discussion section.

ANSWER 17: We have now transferred the text regarding the sample size calculations form Discussion to the Methods section. 

Methods, Selection of study participants, 3rd paragraph:

"Given that the CSS has 36 items, we aimed to minimize bias arising from the number of observations in order to perform a robust confirmatory factor analysis. For this reason we opted for participant-to-item ratio of 25:1 (i.e. minimum sample size 36 x 25 = 900). Although some authors recommend that the participant-to-item ratio be at a minimum 5:1 (29), larger sample sizes could provide more meaningful factor loadings and factors and yield more generalizable results."

18. (L. 373) Please explain what “parallel analysis” refers to and how it “helps to minimize selection bias”.

ANSWER 18: In order to make the study more comprehensible and shorter we previously decided to omit presenting the findings of parallel analysis as well as the discussion about it. Therefore, now we also deleted the sentence in the study strengths regarding the parallel analysis.

---

## [Decision Letter · Decision Letter 1]

12 Oct 2021

Validity and reliability of the Serbian COVID Stress Scales

PONE-D-21-14941R1

Dear Authors,

We’re pleased to inform you that your manuscript has been judged scientifically suitable for publication and will be formally accepted for publication once it meets all outstanding technical requirements.

Kind regards,

Marcel Pikhart

Academic Editor

PLOS ONE

Additional Editor Comments (optional):

Reviewers' comments:

Reviewer's Responses to Questions

**Comments to the Author**

1. If the authors have adequately addressed your comments raised in a previous round of review and you feel that this manuscript is now acceptable for publication, you may indicate that here to bypass the “Comments to the Author” section, enter your conflict of interest statement in the “Confidential to Editor” section, and submit your "Accept" recommendation.

Reviewer #4: All comments have been addressed

2. Is the manuscript technically sound, and do the data support the conclusions?

Reviewer #4: Yes

3. Has the statistical analysis been performed appropriately and rigorously? 

Reviewer #4: Yes

4. Have the authors made all data underlying the findings in their manuscript fully available?

Reviewer #4: Yes

5. Is the manuscript presented in an intelligible fashion and written in standard English?

Reviewer #4: Yes

6. Review Comments to the Author

Reviewer #4: Thanks for having me review this study again. The authors have adequately addressed my comments from the previous round of the review.

7. PLOS authors have the option to publish the peer review history of their article (what does this mean?). If published, this will include your full peer review and any attached files.

Reviewer #4: No

---

## [Editor Report · Acceptance letter]

18 Oct 2021

PONE-D-21-14941R1 

Validity and reliability of the Serbian COVID Stress Scales 

Dear Dr. Gazibara:

I'm pleased to inform you that your manuscript has been deemed suitable for publication in PLOS ONE. Congratulations! Your manuscript is now with our production department. 

Kind regards, 

on behalf of

Dr. Marcel Pikhart 

Academic Editor

PLOS ONE